# Low-Grade Ovarian Serous Adenocarcinoma with Lymph Node Metastasis in Neck

**DOI:** 10.3390/diagnostics11101804

**Published:** 2021-09-29

**Authors:** Shih-Lung Chen, Tsan-Yu Hsieh, Shih-Wei Yang

**Affiliations:** 1Department of Otolaryngology & Head and Neck Surgery, Chang Gung Memorial Hospital, Linkou 333, Taiwan; rlong289@gmail.com; 2School of Medicine, Chang Gung University, Taoyuan 333, Taiwan; 8902059@cgmh.org.tw; 3Department of Pathology, Keelung Change Gung Memorial Hospital, Keelung 204, Taiwan; 4Department of Otolaryngology & Head and Neck Surgery, Chang Gung Memorial Hospital, Keelung 204, Taiwan; 5Department of Otolaryngology & Head and Neck Surgery, New Taipei Municipal Tucheng Hospital, New Taipei City 236, Taiwan

**Keywords:** serous adenocarcinoma, metastasis, ultrasound, fine needle aspiration cytology, positron emission tomography/computed tomography, neck

## Abstract

Low-grade ovarian serous adenocarcinoma is rarely encountered in the neck region. The diagnosis of this rare malignancy entity in the neck is challenging for both clinicians and pathologists. A 53-year-old female with a chief complaint of a right lower neck mass that had been growing for approximately 2 weeks. The ultrasound-guided fine needle aspiration cytology favored malignancy. The positron emission tomography/computed tomography scan revealed the clustered enlarged lymph nodes with increased radioactivity uptake in the right neck level V, and strong radioactivity uptake was also displayed in the right ovarian regions. Pelvis magnetic resonance imaging displayed right adnexal complex mass supporting the ovarian cancer. An en bloc resection of the right neck lymph node was conducted. Ovarian serous adenocarcinoma with metastasis of lymph nodes in the neck was confirmed through histopathological findings. This study reviews the clinical features of low-grade ovarian serous carcinoma metastasizing to lymph nodes in neck. Although very rare, ovarian cancer with neck metastasis should be considered in the differential diagnosis of a neck mass lesion. The clinical staging would be relatively high due to the quiet entity of the cancer.

## 1. Introduction

Supraclavicular metastases from ovarian cancer are rare, and those from the low-grade ovarian serous cancer are extremely rare and frequently misdiagnosed [1]. We report a patient with metastatic low-grade ovarian serous adenocarcinoma to the distant lymph nodes in the neck region diagnosed using ultrasound (US)-guided fine needle aspiration cytology (FNAC), positron emission tomography/computed tomography (PET/CT), and histopathological findings. Furthermore, we compared the clinicopathological characteristics of our case with those of previously published case reports in the literature.

## 2. Case Report

A 53-year-old female came to our department with a chief complaint of a right lower neck mass that had been growing for approximately 2 weeks. The patient had no fever, night sweats, traumatic episodes, past history of surgery, or painful sensations, and neither specific medical disease nor systemic disorder was noted. A physical examination revealed a mass in the right neck level V region. There were no ulcers or mass noted in the oral cavity, oropharynx, nasopharynx, hypopharynx and larynx through fiberoptic endoscopy. Laboratory findings showed a normal white blood cell count without leukocytosis (3500/µL; normal: 3500–11,000/µL). To differentiate this mass lesion, we arranged US-guided FNAC. Target US revealed the ovoid lymph node with a short axis of about 10.3 mm (Figure 1A). FNAC was performed at the site using a 21-gauge needle (Figure 1B). The aspirated cytological findings showed a mixed pus-like lymphocytic fluid with poor-undifferentiated cells supporting malignancy. The PET/CT scan revealed clustered enlarged lymph nodes with increased radioactivity uptake in the right neck level V, and strong radioactivity uptake was also displayed in the right ovarian regions (Figure 2). Furthermore, pelvis magnetic resonance imaging (MRI) displayed right adnexal complex mass supporting the ovarian cancer. An en bloc resection of the right neck lymph node was conducted (Figure 3A,B). The specimen consisted of a greyish and soft mass measuring about 1.4 cm × 1.1 cm × 0.6 cm. Sections of lymph node showed nests of tumor cells arranging in papillary and solid patterns and forming labyrinth structure with slit-like space. Immunohistochemical study showed the tumor cells were folate receptor (+), Wilm’s tumor 1(WT1, +), P53 (+), and estrogen receptor (ER, +) (Figure 4). A diagnosis of metastatic low-grade serous adenocarcinoma was made.

After the diagnosis was confirmed, we further arranged the relevant biochemical blood tests, which revealed elevated levels of CA-125 (307.00 U/mL; normal: <35 U/mL) with normal CA-199, CA153, CEA, and SCC. The primary ovarian cancer metastasizing to regional pelvic and distant neck lymph nodes with clinical staging IV, T3aN2aM1 (American Joint Committee on cancer, 8th version) was impressed. The patient was then referred to an oncologist’s service for the systemic chemotherapy with cisplatin-based regimen. After one year of treatment, the patient was still alive and followed up at the oncologist’s out-patient department.

## 3. Discussion

Ovarian cancer is the seventh most common malignancy, accounting for approximately 3% of all cancers in women, and is the second most common gynecological cancer [2,3]. However, most importantly, it is the leading cause of cancer-related mortality and highest death-to-incidence ratios in women. The five-year survival rate is below 50% due to most patients with ovarian cancer present with advanced disease [4,5].

In ovarian cancer, there are two main categories, including epithelial origin and non-epithelial origin. The category of epithelial origin includes high-grade serous carcinoma, clear cell carcinoma, endometrioid carcinoma, mucinous carcinoma, and low-grade serous carcinoma [6,7]. The low grade-serous carcinoma with precursor lesions in the ovary grows slowly [8,9,10,11]. Ovarian serous adenocarcinoma is a kind of low-grade ovarian serous carcinoma, and it is lethal but rarely metastasizes to neck region [7]. Generally, patients with ovarian serous carcinoma in the distant metastatic lymph nodes usually have a metastatic lesion located outside of the neck region, including the lung, breast, lower digestive tract [2]. The isolated distant metastasis to supra-clavicular neck lymph nodes is unusual. However, in some studies, lymphadenopathy in the neck may be present initially before establishing the final diagnosis of an ovarian tumor [4,12], which was in accordance with our patient. Verbruggen et al. reported that serous tumor with supra-diaphragmatic lymph node involvement can be present in patients, and can even be the presenting symptom [13]. In addition, ovarian serous tumor can also metastasize to brain [14,15].

Four routes of spreading in ovarian cancer have been identified: direct extension, trans-peritoneal route, lymphatic route, and more rarely, through the blood stream [16]. Ovarian cancer spreading through the lymphatic system could result in further extension to the neck lymph nodes. In fact, the sub-peritoneal, infra-diaphragmatic, and diaphragmatic lymphatic vessels are all connected together, and thus the lymphatic fluid route explains neck metastatic lymph nodes in ovarian cancer [3].

In Table 1, the reported cases of low-grade ovarian serous carcinoma with neck metastasis are listed [1,2,4,17,18,19]. The age of diagnosis ranged from 24 to 72 years. Low-grade ovarian serous carcinoma may happen at any age, but most patients occur commonly in the middle age. The systemic evaluation should contain imaging survey and immunohistochemistry [6]. For the imaging studies, CT is the imaging technique of choice, but its capability for detecting distant tumor is limited in cases of tiny metastases. Over the past 20 years, [18F] fluoro-2-deoxy-D-glucose (18F-FDG) positron emission tomography (PET)/computed tomography (CT) have been increasingly used as diagnostic procedures for neck metastasis without an obvious primary site [2]. Several studies have shown that 18F-FDG PET can detect distant tumors in patients suffering from an ovarian carcinoma with equivocal CT findings [4]. PET/CT allows localizing increased 18F-FDG uptake with improved anatomic specificity. Therefore, in gynecological malignancies, it is important to evaluate the patients with PET in terms of distant metastasis. However, it is essential to be aware of the limitations and weakness of PET-CT, including the high rate of false-positive findings, the limited availability of the procedure, the costs of the examination, and the burden to the patient [2]. Infectious and inflammatory diseases are the most frequent cause of false-positive findings. The development of hybrid PET/CT equipment has increased the sensitivity and specificity compared with PET [20].

The main positive markers of immunohistochemistry for ovarian serous adenocarcinoma are ER, CA-125 and WT1. ER and/or progesterone receptor (PR) are positive in 50–83% of ovarian serous carcinoma. WT1 and CA-125 are expressed in the majority of ovarian serous carcinomas, but CA-125 is also expressed in a minority of carcinomas, including those of the breast, endometrium, cervix and lung. The expression of WT1 is often negative in breast, gastrointestinal and pancreatobiliary primaries [2]. 

Low-grade serous carcinoma is considered to evolve in a stepwise fashion from ovarian epithelial inclusions/serous cystadenomas to serous borderline tumors to invasive carcinoma [5]. Pathologically serous adenocarcinoma can contain micro-calcifications, and these calcifications occur in lymph node metastasis. Thus, lymph node metastases should not be confused with old granulomatous disease or calcifications due to previous treatment [4]. Low-grade serous carcinomas exhibit low-grade nuclei with infrequent mitotic figures and are characterized by the presence of mild to moderate nuclear atypia and up to 12 mitoses per 10 high power fields (HPFs) [18,19].

Chemotherapy remains a standard treatment for the disease, and the first-line chemotherapy is a platinum-based regimen (i.e., carboplatin with or without paclitaxel) [1]. However, sometimes resistance would happen [21]. The current consensus is that all patients with advanced or metastatic ovarian cancer should be treated with systemic chemotherapy, radiotherapy and possible optimal debulking surgery [22]. Due to the ovarian serous adenocarcinoma is a hormone-responsive disease, hormone replacement therapy should be avoided in these patients [23]. Additionally, it has been reported that monoclonal antibody bevacizumab in combination with chemotherapy for the disease has promising response rates of 40% [6,24].

In fact, early-stage disease (i.e., disease localized to the ovaries) is associated with an overall 5-year survival rate of over 90%; however, only a small proportion of patients (approximately 15%) present with early disease. On the other hand, five-year survival rates for patients with distant metastatic ovarian cancer are currently estimated to be less than 30% [22]. 

Age has been accepted by consensus as a negative prognostic factor for ovarian cancer survival, especially for those older than 65 years old. Older women have an increased risk of treatment failure and higher rates of recurrences and lower survival rates [25]. The median survival for patients younger than 65 years was at least two years longer, compared to the median survival of patients older than 65 years [26].

Ovarian cancer with neck lymph node metastasis is considered to be distant metastasis, and is classified as stage IV in both International Federation of Gynecology and Obstetrics (FIGO) and AJCC (American Joint Committee on Cancer) stage [12]. Recent evidence suggests that patients suffering from ovarian cancer stage IV with distant lymph nodes as the only distant metastatic site may have a better prognosis than other stage IV patients [27]. 

Our patient was a low-grade ovarian serous adenocarcinoma with an advanced stage due to distant lymph node metastasis. She had no obvious gynecological symptoms and presented the neck mass as the initial manifestation for only 2 weeks. However, the clinical stage was relatively high, and even distant neck lymph node metastasis was displayed due to the quiet entity of cancer. PET/CT showed the increased 18F-FDG uptake in ovarian area and neck region suggesting ovarian cancer with neck metastatic lymph node. Diagnosis was confirmed by histopathologic findings, and high positive CA 125 on immunohistochemistry. In fact, ovarian serous adenocarcinoma belonging to low-grade ovarian serous carcinoma is commonly diagnosed at high stage, and these patients have a poor prognosis, despite this tumor being relatively slow growing and even asymptomatic in the gynecologic field [6]. The clinician should pay more attention to such quiet tumors carefully.

## 4. Conclusions

We demonstrated a patient with rare low-grade ovarian serous adenocarcinoma metastasizing to the lymph nodes in the neck region. US with FNA is the first-line test for assessing a mass before a definitive treatment is launched. PET/CT may show an increased uptake of radioactivity in patient’s neck and pelvic region. Although the diagnosis of metastasis from the ovarian area to neck region is rare, such quiet ovarian tumors with distant metastasis shouldn’t be neglected in the differential diagnosis of lymphadenopathy in the neck.

## Figures and Tables

**Figure 1 diagnostics-11-01804-f001:**
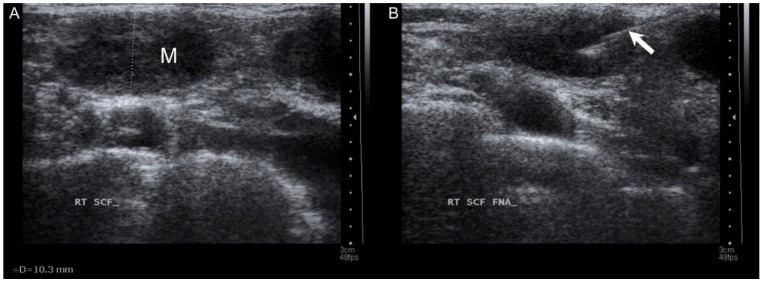
(**A**) Targeted ultrasound revealed an ovoid heterogeneous mass with a short axis of about 10.3 mm. (M: mass); (**B**) fine needle (arrow) aspiration cytology was performed using a 21-gauge needle.

**Figure 2 diagnostics-11-01804-f002:**
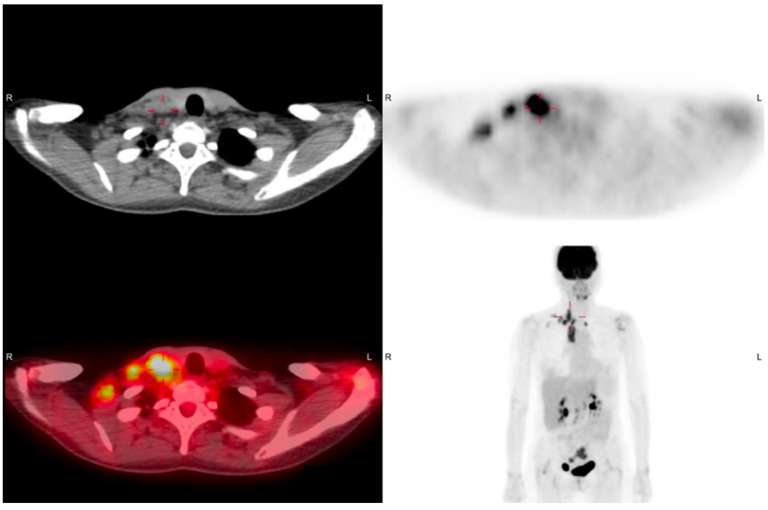
Positron emission tomography/computed tomography revealed the increased radioactivity uptake in the right neck level V and right pelvic adnexa regions.

**Figure 3 diagnostics-11-01804-f003:**
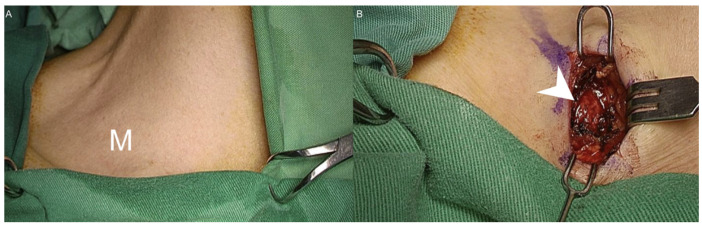
(**A**) A well-defined and fixed mass was noted in the right neck level V area. No erythematous change or local heat over the skin was observed. (M: mass); (**B**) the incision was made above the mass. The left neck mass (arrowhead) was explored and en bloc resection was conducted.

**Figure 4 diagnostics-11-01804-f004:**
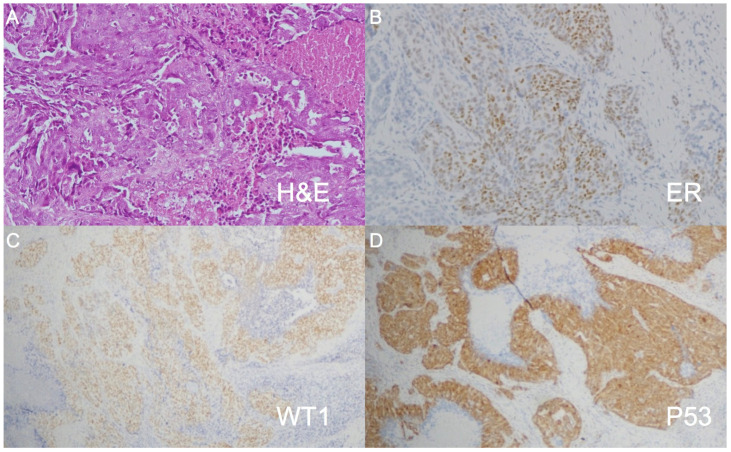
(**A**) The nest of tumor cells displayed papillary and solid patterns and forming labyrinth structure with slit-like space (H&E: hemotoxylin and eosin). (Original magnification ×100); (**B**–**D**) the tumor cells are focally positive for ER (estrogen receptor), diffusely positive for WT1 (Wilm’s tumor 1) and P53. (original magnification ×40).

**Table 1 diagnostics-11-01804-t001:** Reported cases of low-grade ovarian serous tumor with neck metastasis.

The Authors (Year)	Age	Image	Diagnostic Method	Pathological Diagnosis	Cancer Grade	Treatment	Follow-Up, Month
Patel et al. [17] 1999	24	CT	Excision	Serous papillary adenocarcinoma	Low-grade	S	Not available
	32	US, CT	FNAC, excision	Serous papillary adenocarcinoma	Low-grade	C + X	Not available
	44	US, CT	FNAC, excision	Serous papillary adenocarcinoma	Low-grade	S + C	Not available
Malpica et al. [18] 2001	43	Not available	Excision	Serous carcinoma	Low-grade	S + C + X	DOD, 24
	27	Not available	Excision	Serous carcinoma	Low-grade	S + C + X	NED, 72
	25	Not available	Excision	Serous carcinoma	Low-grade	C + X	DOD, 96
	38	Not available	Excision	Serous carcinoma	Low-grade	C + X	NED, 48
Euscher et al. [19] 2004	30	CT	Excision	Serous carcinoma	Low-grade	S + C + X	DOD, 166
	34	CT	Excision	Serous carcinoma	Low-grade	S + C	DOD, 46
Gontier et al. [4] 2006	72	PET/CT	Excision	Serous papillary adenocarcinoma	Low-grade	C	AWD, 24
He et al. [2] 2013	60	US, PET/CT	FNAB, excision	Serous carcinoma	Low-grade	S + C	DOD, 6
Dupret-bories et al. [1] 2015	49	US, CT	FNAC	Serous papillary adenocarcinoma	Low-grade	S + C	NED, 34
Present case, 2021	53	US, PET/CT	FNAC, excision	Serous adenocarcinoma	Low-grade	C	AWD, 2

PET/CT = positron emission tomography/computed tomography; US = ultrasonography; FNAB = fine needle aspiration biopsy; FNAC = fine needle aspiration cytology; DOD = dead of disease; NED = no evidence of disease; AWD = alive with disease; S = surgery; C = chemotherapy; X = radiotherapy.

## Data Availability

Not applicable.

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
