# Peer review of "Low-Grade Ovarian Serous Adenocarcinoma with Lymph Node Metastasis in Neck"

_diagnostics, 2021, doi:10.3390/diagnostics11101804_

Round 1
Reviewer 1 Report
I thank the authors for this interesting case report; please find below my comments:
- I found 13% plagiarism, please improve
- title and text mention head and neck metastasis etc, but the paper deals with neck metastasis mainly. If you want to report also about head metastasis, please consider also other reports of brain metastasis ( like https://doi.org/10.1016/j.inat.2020.100668, 10.1155/2019/2954373. eCollection 2019 etc)
- the follow up is missing in the case illustration section
- remove results section and convert to discussion section
- shorten the discussion (results) section, or divide it in subheadings
Author Response
Comment 1: I thank the authors for this interesting case report; please find below my comments: I found 13% plagiarism, please improve.
Reply 1: Yes, we thank reviewer’s insightful comments. We totally agreed.
We have revised all the quoted paragraphs to prevent any plagiarism. In order to make it easier for reviewers to view these revised paragraphs, we marked the quoted numbers at the end of all quoted paragraphs in red. (From Page 2 Line 73 to Page 4 Line 174).
Comment 2: title and text mention head and neck metastasis etc, but the paper deals with neck metastasis mainly. If you want to report also about head metastasis, please consider also other reports of brain metastasis (like https://doi.org/10.1016/j.inat.2020.100668, 10.1155/2019/2954373. eCollection2019 etc)
Reply 2: Thank for reviewer’s thoughtful comments. We complete appreciated.
As the reviewer said, our article mainly discussed the metastasis of ovarian cancer to the lymph nodes in the neck. The metastasis of the tumor to the head was indeed not discussed in this article. We used “head and neck” is because our otolaryngology department often calls tumors growing in the ear, nose, throat, neck areas, as “head and neck region”.
But we consider the reviewer’s opinion is very valuable, so we remove “head” from the original title and revise it to “Low-grade ovarian serous adenocarcinoma with lymph node metastasis in neck”. In addition, we also revise all “head and neck” to “neck” in the manuscript.
Furthermore, the two researches mentioned by the reviewer are great articles:
doi.org/10.1016/j.inat.2020.100668: Giant cystic brain metastasis from ovarian papillary serous adenocarcinoma: case report and review of the literature
doi.org/10.1155/2019/2954373.eCollection 2019: A rare case of ovarian serous borderline tumor with brain metastasis
Therefore, we use these two articles as our new references [14] and [15]. We also revise the order of all references in the manuscript.
And we add “In addition, ovarian serous tumor can also metastasize to brain [14,15].” on Page 2 Line 91.
New references:
- Giuseppe Emmanuele Umana; Nicola Alberio; Paolo Amico; Anna Maria Lavecchia; Saverio Fagone; Marco Fricia; Giovanni Nicoletti; Salvatore Cicero; Scalia., G. Giant cystic brain metastasis from ovarian papillary serous adenocarcinoma: Case report and review of the literature. Interdisciplinary Neurosurgery 2020, 20, doi:doi.org/10.1016/j.inat.2020.100668.
- Veran-Taguibao, S.; Taguibao, R.A.A.; Gallegos, N.; Farzaneh, T.; Kim, R.; Lin, F.; Lu, D. A Rare Case of Ovarian Serous Borderline Tumor with Brain Metastasis. Case Rep Pathol 2019, 2019, 2954373, doi:10.1155/2019/2954373.
Comment 3: the follow up is missing in the case illustration section
Reply 3: Yes, we appreciate the reviewer’s insightful comments. We completely agreed.
We add “The patient was then referred to an oncologist’s service for the systemic chemotherapy with cisplatin-based regimen. After one year of treatment, the patient was still alive and followed up at oncologist’s out-patient department.” on Page 2, Lines 67-70.
Comment 4: remove results section and convert to discussion section
Reply 4: Yes, we appreciate the reviewer’s thoughtful comments. We completely agree.
We remove results section and convert to discussion section on Page 2 Line 71.
Comment 5: shorten the discussion (results) section, or divide it in subheadings
Reply 5: Yes, we appreciate the reviewer’s thoughtful comments. We completely agree.
We decide to shorten the discussion section, and delete the below paragraphs:
“Another one of the most reliable predictors of survival is the platinum-free interval. Patients who had longer platinum-free intervals also had better survival rates. However, most ovarian cancer patients have high initial response rates, but eventually, recur and require further treatment. Patients with platinum-resistant tumors have poorer response rates to the subsequent chemotherapy regimens and, consequently, shorter overall survival.”
and
“Further, timing of neck lymph node metastasis identification is a key predictor for patients’ survival. Chen et al. found that patients with simultaneous neck lymph node metastasis from ovarian cancer at initial surgical intervention of primary ovarian lesions had better survival outcomes than patients with neck lymph node metastasis discovered at cancer recurrence.”
and
“Other determinants of survival included tumor grade, cellular differentiation, and stage.”
and
“As previous mentioned, ovarian cancer is the leading cancer-related deaths in women, and the high mortality rates can be explained by the delayed diagnosis with more than 75% of affected women being diagnosed at advanced stages of the disease due to lack of or non-specific symptoms, and only 15% is confined to the primary site at the time of diagnosis.”

Reviewer 2 Report
Thank you for the opportunity to review “Low-grade ovarian serous adenocarcinoma with lymph node metastasis in head and neck” (Manuscript ID: diagnostics-1362053) submitted for publication consideration in the MDPI Diagnostics.
In this study the authors report low-grade ovarian serous adenocarcinoma encountered in the head and neck region. The diagnosis was arrived at by utilizing ultrasound-guided fine needle aspiration cytology, positron emission tomography/computed tomography scan, magnetic resonance imaging and finally confirmed by histopathology.
The key message of the case report is that in the differential diagnosis of a neck mass lesion, an ovarian cancer metastasis should be kept in mind.
General comments:
The general format of the paper deployed by the authors includes: 1) Introduction; 2) Case report; 3) Results; and 4) Conclusion.
The authors are requested to go through the journal’s instructions and reorganize the report. Please see below:
Reference: https://www.mdpi.com/journal/diagnostics/instructions
It is observed that the bulk of the results section of the case report (Pages 2-4, Lines 69-178) comprises of discussion. While the actual results reported by the authors is only on page 4 lines 180-186.
Specific comments:
Page 2 Line 46: There is a misspelling of leukocytosis
Page 2 Lines 59-61: The authors report that the ‘Immunohistochemical (IHC) study showed tumor cells were folate receptor (+), Wilm’s tumor 1(WT1,+), P53(+), estrogen receptor (ER, +) (Figure 4) and that the diagnosis of metastatic low-grade serous adenocarcinoma was made’.
The figure 4 is H&E. Although the authors have deployed IHC, they have not included the images in the paper. Is there any specific reason why IHC pictures are not included?
Page 2 to 4 Lines 69-178: Please move this entire section from the subheading ‘results’ section to ‘Discussion’.
Page 2 Lines 76-78: the authors mention ‘The category of epithelial origin includes high-grade serous carcinoma, clear cell carcinoma, endometrioid carcinoma, mucinous carcinoma, and low-grade serous carcinoma.’
The authors have not cited references for this statement.
Page 2 Line 78: The authors state ‘The low grade-serous carcinoma with precursor lesions in the ovary grows slowly’ and have cited Grisham RN (1) and Hatano Y et al. (2). Unless there is a limit to the number of references, the authors attention is drawn to some additional excellent papers on this subject (3-6).
Page 4 Lines 193-194: The authors state ‘US with FNA should be considered as the first-line test for assessing a mass before a definitive treatment is launched.’ The authors may perhaps use another word instead of ‘should’ since ultrasound and FNA are accepted standard diagnostic modalities for mass lesions for over decades.
Figure 1B: the needle inside the lesion needs to be labeled as not every reader of this paper may not be familiar with reading radiological images
Figure 4: histopathology images are Not clear. The sections appear thick, hazy and out of focus. Papillary pattern is not obvious in the sections.
Better resolution images need to be incorporated. High power images must be included. Image magnification needs to be mentioned.
Last but not the least, IHC images must be included to make the paper complete.
- Grisham RN. Low-Grade Serous Carcinoma of the Ovary. Oncology (Williston Park, NY). 2016;30(7):650-2.
- Hatano Y, Hatano K, Tamada M, Morishige KI, Tomita H, Yanai H, et al. A Comprehensive Review of Ovarian Serous Carcinoma. Advances in anatomic pathology. 2019;26(5):329-39.
- McCluggage WG, Judge MJ, Clarke BA, Davidson B, Gilks CB, Hollema H, et al. Data set for reporting of ovary, fallopian tube and primary peritoneal carcinoma: recommendations from the International Collaboration on Cancer Reporting (ICCR). Modern pathology : an official journal of the United States and Canadian Academy of Pathology, Inc. 2015;28(8):1101-22.
- Meinhold-Heerlein I, Fotopoulou C, Harter P, Kurzeder C, Mustea A, Wimberger P, et al. The new WHO classification of ovarian, fallopian tube, and primary peritoneal cancer and its clinical implications. Arch Gynecol Obstet. 2016;293(4):695-700.
- Duska LR, Kohn EC. The new classifications of ovarian, fallopian tube, and primary peritoneal cancer and their clinical implications. Annals of oncology : official journal of the European Society for Medical Oncology. 2017;28(suppl_8):viii8-viii12.
- Slomovitz B, Gourley C, Carey MS, Malpica A, Shih IM, Huntsman D, et al. Low-grade serous ovarian cancer: State of the science. Gynecologic oncology. 2020;156(3):715-25.

Author Response
Revision of “Low-grade ovarian serous adenocarcinoma with lymph node metastasis in the neck”
Reviewer 2
Comment 1
Thank you for the opportunity to review “Low-grade ovarian serous adenocarcinoma with lymph node metastasis in head and neck” (Manuscript ID: diagnostics-1362053) submitted for publication consideration in the MDPI Diagnostics.
In this study the authors report low-grade ovarian serous adenocarcinoma encountered in the head and neck region. The diagnosis was arrived at by utilizing ultrasound-guided fine needle aspiration cytology, positron emission tomography/computed tomography scan, magnetic resonance imaging and finally confirmed by histopathology. The key message of the case report is that in the differential diagnosis of a neck mass lesion, an ovarian cancer metastasis should be kept in mind.
General comments: The general format of the paper deployed by the authors includes:
(1) Introduction; (2) Case report; (3) Results; and (4) Conclusion.
The authors are requested to go through the journal’s instructions and reorganize the report. Please see below: Reference: https://www.mdpi.com/journal/diagnostics/instructions
Reply 1: Yes, we appreciate the reviewer’s thoughtful comments. We completely agreed.
We revise our manuscript as “(1) Introduction; (2) Case report; (3) Discussion (Page 2 Line 71); and (4) Conclusion.” to meet the journal’s instructions.
Comment 2
It is observed that the bulk of the results section of the case report (Pages 2-4, Lines 69-178) comprises of discussion. While the actual results reported by the authors is only on page 4 lines 180-186.
Reply 2: Thank for reviewer’s thoughtful comments. We complete appreciated.
We revise the original “Results” section to “Discussion” section. The revised “Discussion” section is from Page 2 Line 71 to Page 4 Line 174, which meet the author’s instructions of the journal.
Comment 3
Specific comments: Page 2 Line 46: There is a misspelling of leukocytosis.
Reply 3: Thank for reviewer’s detailed review. We completely appreciate.
We correct “leukocytosius” to “leukocytosis” on Page 2 Line 47. Thank the reviewer 2.
Comment 4
Page 2 Lines 59-61: The authors report that the ‘Immunohistochemical (IHC) study showed tumor cells were folate receptor (+), Wilm’s tumor 1(WT1,+), P53(+), estrogen receptor (ER, +) (Figure 4) and that the diagnosis of metastatic low-grade serous adenocarcinoma was made’.
The figure 4 is H&E. Although the authors have deployed IHC, they have not included the images in the paper. Is there any specific reason why IHC pictures are not included?
Reply 4: Thank for reviewer’s insightful comments. We complete appreciated.
We revise Figure 4A and added pictures of IHC staining with ER, WT1 and P53 as Figure 4B-D. We upload the new Figure 4A-D and revise the new figure legends on Page 6 Lines 202-205 as below:
Figure 4A. The nest of tumor cells displayed papillary and solid patterns and forming labyrinth structure with slit-like space (original magnification Í100).
Figure 4B-D. The tumor cells are focally positive for ER (estrogen receptor), diffusely positive for WT1 and P53. (original magnification Í40).
- We tried to find the picture with folate receptor (+), but the picture of file was destroyed.
Comment 5
Page 2 to 4 Lines 69-178: Please move this entire section from the subheading ‘results’ section to ‘Discussion’.
Reply 5: Thank for reviewer’s thoughtful comments. We complete appreciated.
We revise the original “Results” section to “Discussion” section. The revised “Discussion” section is from Page 2 Line 71 to Page 4 Line 174.
Comment 6
Page 2 Lines 76-78: the authors mention ‘The category of epithelial origin includes high-grade serous carcinoma, clear cell carcinoma, endometrioid carcinoma, mucinous carcinoma, and low-grade serous carcinoma.’ The authors have not cited references for this statement.
Reply 6 Thank for reviewer’s thoughtful comments. We complete appreciated.
The references are
[6] Grisham, R.N. Low-Grade Serous Carcinoma of the Ovary. Oncology (Williston Park) 2016, 30, 650-652.
and
[7] Hatano, Y.; Hatano, K.; Tamada, M.; Morishige, K.I.; Tomita, H.; Yanai, H.; Hara, A. A Comprehensive Review of Ovarian Serous Carcinoma. Adv Anat Pathol 2019, 26, 329-339, doi:10.1097/PAP.0000000000000243.
We add the cited refenced at the end of the sentence as below:
“The category of epithelial origin includes high-grade serous carcinoma, clear cell carcinoma, endometrioid carcinoma, mucinous carcinoma, and low-grade serous carcinoma [6,7].” on Page 2 Lines 79-81.
Comment 7
Page 2 Line 78: The authors state ‘The low grade-serous carcinoma with precursor lesions in the ovary grows slowly’ and have cited Grisham RN (1) and Hatano Y et al. (2). Unless there is a limit to the number of references, the authors attention is drawn to some additional excellent papers on this subject (3-6).
Reply 7 Thank for reviewer’s thoughtful comments. We complete appreciated.
We revise our cited reference and add these excellent papers (3-6) as our new references [8-11].
We add the new reference at the end of the sentence “The low grade-serous carcinoma with precursor lesions in the ovary grows slowly [8-11].” on Page 2 Lines 81-82. We also revised the order of all references in the manuscript. Thank reviewer again, we really benefited and learned a lot from these new articles.
References:
(1). Grisham RN. Low-Grade Serous Carcinoma of the Ovary. Oncology (Williston Park, NY). 2016;30(7):650-2.
(2). Hatano Y, Hatano K, Tamada M, Morishige KI, Tomita H, Yanai H, et al. A Comprehensive Review of Ovarian Serous Carcinoma. Advances in anatomic pathology. 2019;26(5):329-39.
(3). McCluggage WG, Judge MJ, Clarke BA, Davidson B, Gilks CB, Hollema H, et al. Data set for reporting of ovary, fallopian tube and primary peritoneal carcinoma: recommendations from the International Collaboration on Cancer Reporting (ICCR). Modern pathology : an official journal of the United States and Canadian Academy of Pathology, Inc. 2015;28(8):1101-22.
(4). Meinhold-Heerlein I, Fotopoulou C, Harter P, Kurzeder C, Mustea A, Wimberger P, et al. The new WHO classification of ovarian, fallopian tube, and primary peritoneal cancer and its clinical implications. Arch Gynecol Obstet. 2016;293(4):695-700.
(5). Duska LR, Kohn EC. The new classifications of ovarian, fallopian tube, and primary peritoneal cancer and their clinical implications. Annals of oncology : official journal of the European Society for Medical Oncology. 2017;28(suppl_8):viii8-viii12.
(6). Slomovitz B, Gourley C, Carey MS, Malpica A, Shih IM, Huntsman D, et al. Low-grade serous ovarian cancer: State of the science. Gynecologic oncology. 2020;156(3):715-25.
Comment 8
Page 4 Lines 193-194: The authors state ‘US with FNA should be considered as the first-line test for assessing a mass before a definitive treatment is launched.’ The authors may perhaps use another word instead of ‘should’ since ultrasound and FNA are accepted standard diagnostic modalities for mass lesions for over decades.
Reply 8: Thank for reviewer’s thoughtful comments. We complete agreed.
As the reviewer said, ultrasound and FNA are accepted standard diagnostic modalities for mass lesions for over decades. Therefore, we removed the “should”, and revised the sentence to “US with FNA is the first-line test for assessing a mass before a definitive treatment is launched.” on Page 4 Lines 177-178.
Comment 9
Figure 1B: the needle inside the lesion needs to be labeled as not every reader of this paper may not be familiar with reading radiological images.
Reply 9: Thank for reviewer’s insightful comments. We complete agreed.
We add the “arrow” to label the needle inside the lesion in Figure 1B, and we revise the legend of Figure 1B as “Fine needle (arrow) aspiration cytology was performed using a 21-gauge needle” on Page 5 Line 191.
Comment 10
Figure 4: histopathology images are Not clear. The sections appear thick, hazy and out of focus. Papillary pattern is not obvious in the sections. Better resolution images need to be incorporated. High power images must be included. Image magnification needs to be mentioned. Last but not the least, IHC images must be included to make the paper complete.
Reply 10: Thank for reviewer’s insightful comments. We complete appreciated.
As reviewer said, the original Figure 4A appear thick, hazy and out of focus. Papillary pattern is not obvious in the sections. We completely agreed that IHC images must be included to make the paper complete.
Therefore, we revise the original Figure 4A and add pictures of IHC staining with ER, WT1 and P53 as Figure 4B-D. We upload the new Figure 4A-D and revise the figure legends on Page 6 Lines 202-205 as below:
Figure 4A. The nest of tumor cells displayed papillary and solid patterns and forming labyrinth structure with slit-like space (original magnification Í100).
Figure 4B-D. The tumor cells are focally positive for ER (estrogen receptor), diffusely positive for WT1 and P53. (original magnification Í40).

Round 2
Reviewer 1 Report
The authors addressed the requested modification and the paper is much improved. Congrats.
Reviewer 2 Report
Thank you for revising the manuscript. It reads much better. Congratulations.